# Deep learning for genomics using Janggu

Wolfgang Kopp [1✉], Remo Monti [1,2], Annalaura Tamburrini[1,3], Uwe Ohler [1,4] & Altuna Akalin [1✉]

In recent years, numerous applications have demonstrated the potential of deep learning for an improved understanding of biological processes. However, most deep learning tools developed so far are designed to address a specific question on a fixed dataset and/or by a fixed model architecture. Here we present Janggu, a python library facilitates deep learning for genomics applications, aiming to ease data acquisition and model evaluation. Among its key features are special dataset objects, which form a unified and flexible data acquisition and pre-processing framework for genomics data that enables streamlining of future research applications through reusable components. Through a numpy-like interface, these dataset objects are directly compatible with popular deep learning libraries, including keras or pytorch. Janggu offers the possibility to visualize predictions as genomic tracks or by exporting them to the bigWig format as well as utilities for keras-based models. We illustrate the functionality of Janggu on several deep learning genomics applications. First, we evaluate different model topologies for the task of predicting binding sites for the transcription factor JunD. Second, we demonstrate the framework on published models for predicting chromatin effects. Third, we show that promoter usage measured by CAGE can be predicted using DNase hypersensitivity, histone modifications and DNA sequence features. We improve the performance of these models due to a novel feature in Janggu that allows us to include high-order sequence features. We believe that Janggu will help to significantly reduce repetitive programming overhead for deep learning applications in genomics, and will enable computational biologists to rapidly assess biological hypotheses.

[1] Berlin Institute for Medical Systems Biology, Max Delbrueck Center for Molecular Medicine, 10115 Berlin, Germany. [2] Digital Health Machine Learning, Hasso Plattner Institute, University of Potsdam, 14482 Potsdam, Germany. [3] Department of Biology, Centro di Bioinformatica Molecolare, University of Rome 'Tor Vergata', 00133 Rome, Italy. [4] Department of Biology, Humboldt University, 10115 Berlin, Germany. ✉email: wolfgang.kopp@mdc-berlin.de; altuna.akalin@mdc-berlin.de

The recent explosive growth of biological data, particularly in the field of regulatory genomics, has continuously improved our understanding about regulatory mechanism in cell biology[1]. Meanwhile, the remarkable success of deep neural networks in other areas, including computer vision, has attracted attention in computational biology as well. Deep learning methods are particularly attractive in this case, as they promise to extract knowledge in a data-driven fashion from large datasets while requiring limited domain expertise[2]. Since their introduction[3,4], deep learning methods have dominated computational modeling strategies in genomics where they are now routinely used to address a variety of questions ranging from the understanding of protein binding from DNA sequences[3], epigenetic modifications[4–6], predicting gene-expression from epigenetic marks[7], or predicting the methylation state of single cells[8].

Despite the success of these numerous deep learning solutions and tools, their broad adaptation by the bioinformatics community has been limited. This is partially due to the low flexibility of the published methods to adapt to new data, which often requires a considerable engineering effort. This situation illustrates a need for software frameworks that allow for a fast turnover when it comes to addressing new hypotheses, integrating new datasets, or experimenting with new neural network architectures.

In fact, several recent packages, including pysster[9], kipoi[10] and selene[11], have been proposed to tackle this issue on different levels. However, they are limited in their expressiveness and flexibility due to a restricted programming interface or supporting only specific types of models (e.g. by support of single-modal as opposed to multi-modal models that use DNA or protein sequences as input)[9,11], a focus on reproducibility and reusability of trained models but not the entire training process[10], or the adoption of a specific neural network library through a tight integration[11]. All of them have in common that the support of different data types beyond sequence is limited.

To address some of these shortcomings, we present Janggu, a python library for deep learning in genomics, which is named after a hourglass-shaped Korean percussion instrument whose two ends reflect the two ends of a deep learning application, namely data acquisition and evaluation. The library supports flexible prototyping of neural network models by separating the pre-processing and dataset specification from the modeling part. Accordingly, Janggu offers dedicated genomics dataset objects. These objects provide easy access and pre-processing capabilities to fetch data from common file formats, including FASTA, BAM, bigWig, and BED files (see Fig. 1), and they are directly compatible with commonly used machine learning libraries, such as keras, pytorch or scikit-learn. In this way, they effectively bridge the gap between commonly used file formats in genomics and the python data format that is understood by the deep learning libraries. The dataset objects can be easily reused for different applications, and they place no restriction on the model architecture to be used with. A key advantage of establishing reusable and well-tested dataset components is to allow for a faster turnaround when it comes to setting up deep learning models and increased flexibility for addressing a range of questions in genomics. As a consequence, we expect significant reductions in repetitive software engineering aspects that are usually associated with the pre-processing steps. We illustrate Janggu on three use cases: (1) predicting transcription factor binding of JunD, (2) using and improving published deep learning architectures, and (3) predicting normalized CAGE-tag counts at promoters. In these examples, different data formats are consumed, including FASTA, bigWig, BAM, and narrowPeak files. Here, we also make use of Janggu's ability of using higher order sequence features (see Hallmarks), and show that this leads to significant performance improvements.

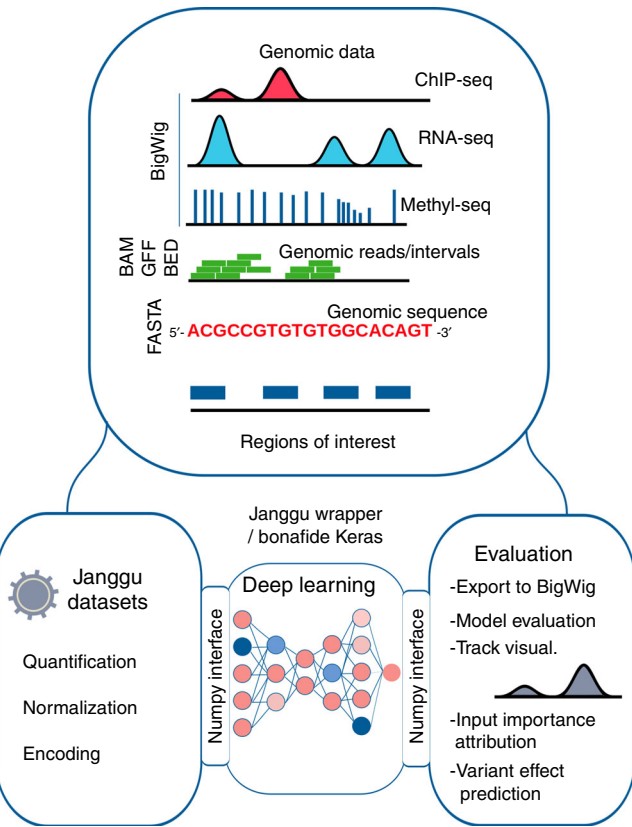

**Fig. 1 Janggu schematic overview.** Janggu helps with data aquisition and evaluation of deep learning models in genomics. Data can be loaded from various standard genomics file formats, including FASTA, BED, BAM, and bigWig. The output predictions can be converted back to coverage tracks and exported to bigWig files.

## Results

**Hallmarks of Janggu.** Janggu offers two special dataset classes: Bioseq and Cover, which can be used to conveniently load genomics data from a range of common file formats, including FASTA, BAM, bigWig, or BED files. Biological sequences (e.g. from the reference genome) and coverage information (e.g. from BAM, bigWig or BED files) are loaded for user-specified regions of interest (ROI), which are provided in BED-like format. Since Bioseq and Cover both mimic a minimal numpy interface, the objects may be directly consumed using e.g. keras or scikit-learn.

Bioseq and Cover provide a range of options, including the binsize, step size, or flanking regions for traversing the ROI. The data may be stored in different ways, including as ordinary numpy arrays, as sparse arrays or in hdf5 format, which allow the user to balance the trade-off between speed and memory footprint of the application. A built-in caching mechanism helps to save processing time by reusing previously generated datasets. This mechanism automatically detects if the data have changed and needs to be reloaded.

Furthermore, Cover and Bioseq expose dataset-specific options. For instance, coverage tracks can be loaded at different resolution (e.g. base-pair or 50-bp resolution) and they can be subjected to various normalization and transformation steps, including TPM normalization or log transformation. The possibility to convert raw numpy array format to coverage objects allows to exported model predictions as bigWig format or visualize them via the built-in plotGenomeTrack function. Bioseq also enables the user to work with both DNA and protein sequences. Here, sequences

can be one-hot encoded using higher order sequence features, allowing the models to learn e.g. di- or tri-mer based motifs.

Additionally, Janggu offers a number of features that are based on a keras integration, including (1) specific keras layers e.g. for scanning both DNA strands or a model wrapper that enables (2) exporting of commonly used performance metrics directly within the framework (e.g. area under the precision-recall curve), (3) input feature importance attribution via integrated gradients[12], and (4) evaluating variant effect for single nucleotide variants taking advantage of the higher order sequence representation.

A schematic overview is illustrated in Fig. 1. Further details on its functionality are available in the documentation at https://janggu.readthedocs.io.

**Prediction of JunD binding**. To showcase different Janggu functionalities, we defined three example problems to solve by utilizing our framework. We start by predicting the binding events of the transcription factor JunD. JunD binding sites exhibit strong interdependence between nucleotide positions[13], suggesting that it might be beneficial to take the higher order sequence composition directly into account. To this end, we introduce a higher order one-hot encoding of the DNA sequence that captures e.g. di- or tri-nucleotide based motifs, which is available with the Bioseq object. For example, for a sequence of length $N$, the di-nucleotide one-hot encoding corresponds to a $16 \times N - 1$ matrix, where each column contains a single *one* in the row that is associated with the di-nucleotide at that position. We shall refer to mono-, di- and tri-nucleotide encoding as order one, two and three, respectively. In contrast to mono-nucleotide input features, higher order features directly capture correlations between neighboring nucleotides.

For JunD target predictions, we observe a significant improvement in area under the precision-recall curve (auPRC) on the test set when using the higher order sequence encoding compared to the mono-nucleotide encoding (see Fig. 2a, red). The median performance gain across five runs amounts to $\Delta$auPRC = 8.3% between order 2 and 1, as well as $\Delta$auPRC = 9.3% between order 3 and 1.

While the use of higher order sequence features uncovers useful information for interpreting the human genome, the larger input and parameter space might make the model prone to overfitting, depending on the amount of data and the model complexity. We tested whether dropout on the input layer, which randomly sets a subset of ones in the one-hot encoding to zeros, would improve model generalization[14]. Using dropout on the input layer should also largely preserve the information content of the sequence encoding, as the representation of higher orders is inherently redundant due to overlapping neighboring bases.

In line with our expectations, dropout leads to a slight further performance improvement for tri-nucleotide-based sequence encoding. On the other hand, for mono-nucleotide-based encoding we observe a performance decrease. We observe slightly worse performance also when using di-nucleotide-based encoding, suggesting that the model is over-regularized with the addition of dropout. However, dropout might still be a relevant option for the di-nucleotide based encoding if the amount of data is relatively limited (see Fig. 2a).

As many other transcription factors, JunD sites are predominately localized in accessible regions in the genome, for instance as assayed via DNase-seq[15]. To investigate this further, we set out to predict JunD binding from the raw DNase cleavage coverage profile in 50 bp resolution extracted from BAM files of two independent replicates simultaneously (from ENCODE and ROADMAP, see Methods).

Raw read coverage obtained from BAM files is inherently biased, e.g. due to differences in sequencing depths, etc., which requires normalization in order to achieve comparability between experiments. As a complementary approach, data augmentation has been shown to improve generalization of neural networks by increasing the amount of data by additional perturbed examples of the original data points[16]. Accordingly, we compare TPM normalization and Z score normalization of log(count + 1) in combination with data augmentation by flipping the 5' to 3' orientation of the coverage tracks. To test the effectiveness of normalization and data augmentation, we swapped the input DNase experiments from ENCODE and ROADMAP between training and test phase. The more adequate the normalization, the higher we anticipate the performance to be on the test set.

We find that both TPM and Z score after log(count + 1) transformation lead to improved performance compared to applying no normalization, with the Z score after log(count + 1) transformation yielding the best results (see Fig. 2b). The additional application of data augmentation tends to slightly improve the performance for predicting JunD binding from DNase-seq (see Fig. 2b).

Next, we build a combined model for predicting JunD binding based on the DNA sequence and DNase coverage tracks. To that end, we used the same initial layers as for the order-3 DNA model and the DNase-specific models using Z score after log(count + 1)-normalization with orientation flipping. We removed their output layers, concatenated the top most hidden layers, and added a new sigmoid output layer. We trained the joint model from scratch using randomly initialized weights for all layers and found that its performance significantly exceeded the performance of the individual DNA and DNase submodels, indicating that both ingredients contributed substantially to the predictive performance (compare Fig. 2a–c).

As a means to inspect the plausibility of the results apart from summary performance metrics (e.g. auPRC), Janggu features a built-in genome track plotting functionality that can be used to visualize the agreement between predicted and known binding sites, or the relationship between the predictions and the input coverage signal for a selected region (Fig. 2d). Input importance attribution using integrated gradients[12] additionally highlights the relevance of sequence features for the prediction, which in the case of the JunD prediction task reveals sequence patches reminiscent of the known JunD binding motif (with the canonical sequence motif TGACTCA) close to the center of the predicted peak (Fig. 2d).

**Predicting chromatin profiles from genomic sequences**. Predicting the function of non-coding sequences in the genome remains a challenge. In order to address this challenge and assess the functional relevance of non-coding sequences and sequence variants, multiple deep learning based models have been proposed. These models learn the genomic sequence features that give rise to chromatin profiles such as transcription binding sites, histone modification signals or DNase hypersensitive sites. We adopted two published neural network models that are designed for this purpose, which have been termed DeepSEA and DanQ[4,17]. We rebuilt these models using the Janggu framework to predict the presence (or absence) of 919 genomic and epigenetic features, including DNase hypersensitive sites, transcription factor binding events and histone modification marks, from the genomic DNA sequence. To that end, we gathered and reprocessed the same features, making use of Janggu's pre-processing functionality[4] (see Methods). Both published models were adapted to scan both DNA strands simultaneously in the first layer rather than just the forward strand as this leads to slight

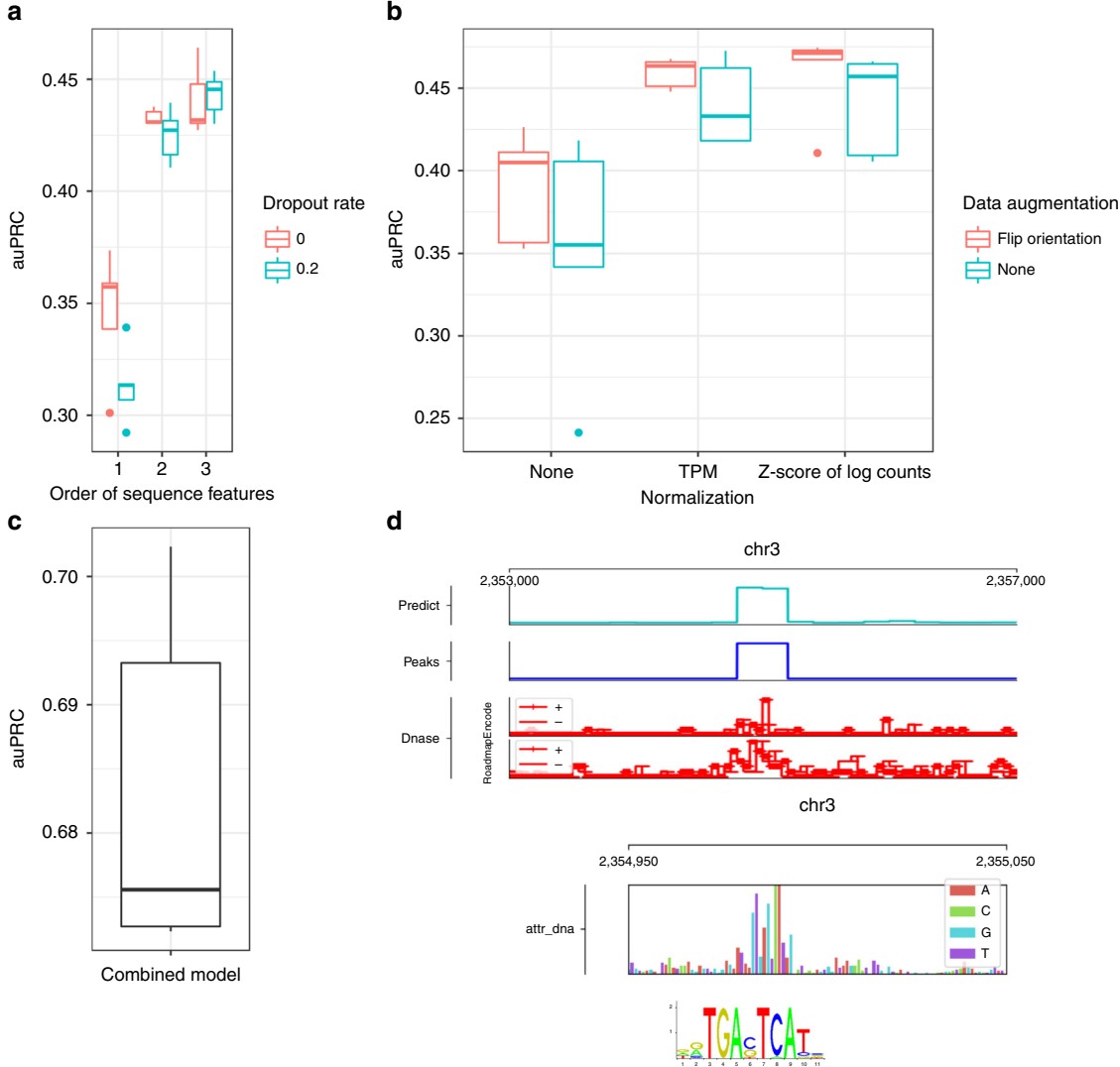

**Fig. 2 Performance evaluation of JunD prediction. a** Performance comparison of different one-hot encoding orders enabled by Janggu's Bioseq object. Order 1, 2, and 3 correspond to mono-, di- and tri-nucleotide based one-hot encoding, respectively. Each model was trained from scratch for five times using random initial weights. Boxes represent quartiles Q1 (25% quantile), Q2 (median) and Q3 (75% quantile); whiskers comprise data points that are within 1.5 x IQR (inter-quartile region) of the boxes. **b** Performance comparison of different normalization and data augmentation strategies applied to the read counts from the BAM files. We compared (1) No normalization (None), (2) TPM normalization, and (3) Z score of log(count + 1) which are optionally available via the Cover object. Moreover, data augmentation consisted of (1) no augmentation (None) or (2) randomly flipping 5' to 3' orientations which was employed by Janggu's dataset wrappers. Each model was trained from scratch for five times using random initial weights. Boxplots are defined as in (**a**). **c** Performance for JunD prediction for the combined model that takes DNA and DNase coverage into account. Each model was trained from scratch for five times using random initial weights. Boxplots are defined as in (**a**). **d** Example of a JunD binding site. The top-most panel shows predicted, true JunD binding site as well as the input DNase coverage around the peak. Underneath integrated gradients further highlights the importance of a site reminiscent of the known JunD motif (Jaspar motif: MA091.1).

performance improvements (see DnaConv2D layer, Janggu documentation). Then we assessed the performance of the different models by considering different context window sizes (500 bp, 1000 bp, and 2000 bp) as well as different one-hot encoding representations (based on mono-, di- and tri-nucleotide content).

First, in agreement with Quang and Xie[17], we find that the DanQ model consistently outperforms the DeepSEA model (as measured by auPRC) in our benchmark analysis regardless of the context window size, one-hot encoding representation and features type (e.g. histone modification, DNase hypersensitive sites and TF binding sites) (see Supplementary Fig. 1). Second, in line with previous reports[4,6], we find the performance for histone modifications and histone modifiers (e.g. Ezh2, Suz12, etc.) to benefit from extending the context window sizes (see Fig. 3a and

Supplementary Fig. 1). By contrast, elongating the context window yields similar performance for accessible sites and transcription factor binding-related features.

Third, higher order sequence encoding influences predictions for histone modification, DNase and TF binding associated features differently. For histone modification predictions we observe mildly improved performances for higher order over mono-nucleotide based one-hot encoding with a median improvement of approximately 1% auPRC across all marks. By contrast, the DNase accessibility and transcription factor binding we observe a median increase in auPRC by 4.1% and 3.3% (see Fig. 3b, c). While most transcription factor binding predictions are influenced mildly, there exist a number of TFs for which substantial improvements are obtained (see Fig. 3b, c). Among

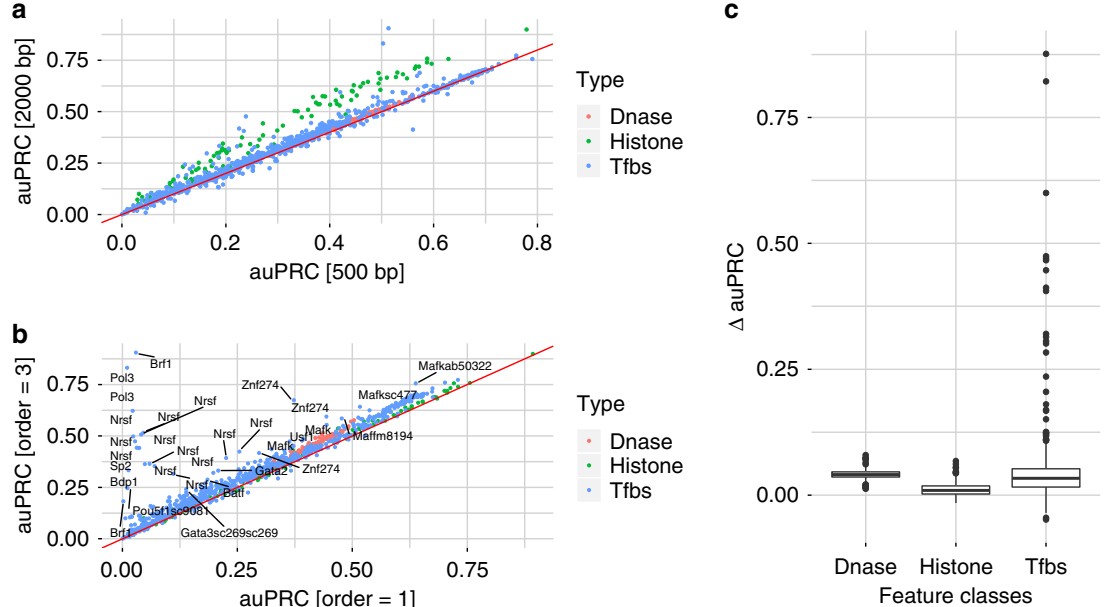

**Fig. 3 Comparison of DanQ model variants. a** auPRC comparison for the context window sizes 500 bp and 2000 bp for tri-nucleotide based sequence encoding. The mark color indicates the feature types: DNase hypersensitive sites, histone modifications and transcription factor binding assays. **b** auPRC comparison for tri- and mono-nucleotide based sequence encoding for a context window of 2000 bp. Color coding as above. **c** Differences in auPRC between tri- and mono-nucleotides for DNase accessibility, histone modifications and transcription factor binding, respectively. $\Delta auPRC > 0$ indicates improved performance for the tri-nucleotide based encoding. Dnase, histone modification and TF features comprise $n = 125$, $n = 104$, and $n = 690$ samples, respectively. Boxes represent quartiles Q1 (25% quantile), Q2 (median), and Q3 (75% quantile); whiskers comprise data points that are within 1.5 x IQR (inter-quartile region) of the boxes.

the most prominent performance improvements are found for Nrsf, Pol3, Sp2, etc. (see Fig. 3b). On the other hand, we observe less variability for the predictions of the DNase accessibility features.

**Predicting CAGE signal at promoters**. Finally, we used Janggu for the prediction of promoter usage of protein coding genes. Specifically, we built a regression application for predicting the normalized CAGE-tag counts at promoters of protein coding genes based on chromatin features (DNase hypersensitivity and H3K4me3 signal) and/or DNA sequence features. Due to the limited amount of data for this task, we pursue a per-chromosome cross-validation strategy (see Methods).

We trained a model using only the DNA sequence as input with different one-hot encoding orders. Consistent with the previous use cases, we observe that the use of higher order sequence features markedly improves the performance from 0.533 (average Pearson's correlation) to 0.559 and 0.585 for mono-nucleotide features compared to di- and tri-nucleotide based features, respectively (see Table 1). Predictions from chromatin features alone yield a substantially higher average Pearson's correlation of 0.777 compared to using the DNA sequence models (see Table 1).

Similar to the previous sections, we concatenate the individual top most hidden layers and add new output layer to form a joint DNA and chromatin model. Consistent with our results from the JunD prediction, the Pearson's correlation between observed and predicted values increases for the combined model (see Table 1 and Fig. 4), even though the difference seems to be subtle in this scenario. The results also show that chromatin features vastly dominate the prediction accuracy compared to the contribution of the DNA sequence. This is expected due to the fact that the DNA sequence features are collected only from a narrow window around the promoter. On the other hand, the chromatin features

**Table 1 Average Pearson's correlation across the cross-validation runs.**

| Model | DNA order | Mean Pearson's corr. | Stand. error |
|---|---|---|---|
| Chromatin only | – | 0.777 | $2.97 \times 10^{-5}$ |
| DNA only | 1 | 0.533 | $4.38 \times 10^{-3}$ |
| DNA only | 2 | 0.559 | $8.40 \times 10^{-3}$ |
| DNA only | 3 | 0.585 | $6.47 \times 10^{-3}$ |
| DNA & Chromatin | 1 | 0.775 | $6.01 \times 10^{-4}$ |
| DNA & Chromatin | 2 | 0.783 | $5.13 \times 10^{-4}$ |
| DNA & Chromatin | 3 | 0.784 | $5.15 \times 10^{-4}$ |

Correlation was measured between observed and predicted normalized CAGE-counts. The models use DNA or Chromatin (DNase and H3K4me3) either separately or simultaneously as input. Different one-hot encoding orders were compared to represent DNA sequences.

reflect activation not only due to the local context, but also due to indirect activation from distal regulatory elements, e.g. enhancers.

## Discussion

We present Janggu, a python library that facilitates deep learning in genomics. The library includes dataset objects that manage the extraction and transformation of coverage information as well as fetching biological sequence directly from a range of commonly used file types, including FASTA, BAM, or bigWig. These dataset objects may be consumed directly with numpy-compatible deep learning libraries, e.g. keras, due to the fact that they mimic a minimal numpy interface, which in turn reduces the software engineering effort concerning the data acquisition for a range of deep learning applications in genomics. Model prediction or features can be inspected using a built-in genome browser or they

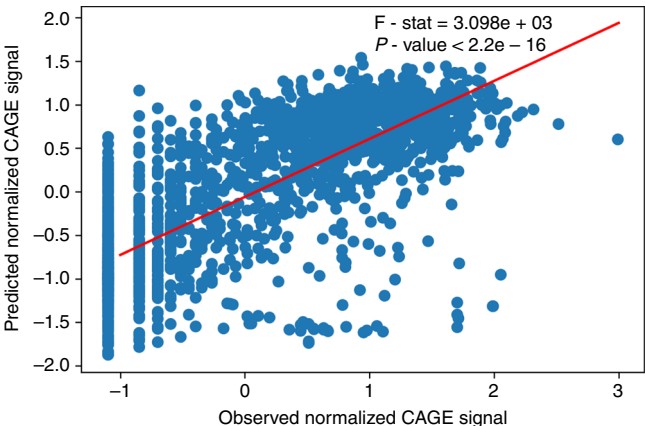

**Fig. 4 Agreement between predicted and observed CAGE signal.** The example illustrates the agreement between predicted and observed CAGE signal on the test chromosome for the joint DNA-chromatin model. The DNA was represented as tri-nucleotide based one-hot encoding. A linear regression (red line) was fitted in order to test the agreement between predicted and observed CAGE signal. Significance of the explained variability was tested using an F-test (P-value < $2.2 \times 10^{-16}$; F-stat = $3.098 \times 10^3$, one-sided).

may be exported to bigWig files for further investigation. Added support for keras models enables input feature importance analysis using integrated gradient and variant effects may assessed for a given VCF format file as well as monitoring of training and performance evaluation.

We have demonstrated the use of Janggu for three case studies that (1) utilize different data types from a range of commonly file formats (FASTA, BAM, bigWig, BED, and GFF) in single- and multi-modal modeling settings alike (e.g. which use DNA sequences or coverage or some combination as input), (2) require different pre-processing and data augmentation strategies, (3) show the advantage of one-hot encoding of higher order sequence features (representing mono-, di-, and tri-nucleotide sequences), and (4) for a classification and regression task (JunD prediction and published models) and a regression task (CAGE-signal prediction). This illustrates our tool is readily applicable and flexible to address a range of questions allowing users to more effectively concentrate on testing biological hypothesis.

Throughout the use cases we confirmed that higher order sequence features improve deep learning models. This is in particular the case for describing a subset of transcription factor binding events, because they simultaneously convey information about the DNA sequence and shape[18]. While, higher order sequence models have been demonstrated to outperform commonly used position weight matrix-based binding models[19], they have received less attention by the deep learning community in genomics. Even though mono-nucleotide-based one-hot encoding approach captures higher order sequence features to some extent by combining the sequence information in a complicated way through e.g. multiple convolutional layers[13], our results demonstrate that it is more effective to capture correlations between neighboring nucleotides at the initial layer, rather than to defer this responsibility to subsequent convolutional layers. Performance improvements of the due to the higher order sequence encoding potentially translate into improved variant effect predictions, at least for a subset of TFs, because the variant effect predictions depend directly on the model predictions. Therefore, Janggu exposes variant effect prediction functionality, similar as Kipoi and Selene[10,11], which allows to make use of the higher order sequence encoding.

## Methods

**Dataset and evaluation for JunD prediction**. We downloaded JunD peaks (ENCFF446WOD, conservative IDR thresholded peaks, narrowPeak format), and raw DNase-seq data (ENCFF546PJU, Stam. Lab, ENCODE; ENCFF059BEU Stam. Lab, ROADMAP, bam-format) for human embryonic stem cells (H1-hesc) from the encodeproject.org and the hg38 reference genome. Alignment indices were built with samtools. Blacklisted regions for hg38 were obtained from http://mitra.stanford.edu/kundaje/akundaje/release/blacklists/hg38-human/hg38.blacklist.bed.gz and removed using bedtools.

We defined all chromosomes as training chromosomes except for chr2 and chr3 which are used as validation and test chromosomes, respectively. The region of interest was defined as the union of all JunD peaks extended by 10 kb with a binning of 200 bp. Each 200 bp-bin is considered a positive labels if it overlaps with a JunD peak. Otherwise it is considered a negative example. For the DNA sequence, we further extended the context window by ±150 bp leading to a total window size of 500 bp. Similarly, for the DNase signal, we extracted the coverage in 50 bp resolution adding a flanking region of ±450 bp to each 200 bp window which leads to a total input window size of 1100 bp.

The training and evaluation labels were loaded into a Cover object using the create_from_bed method, the DNA sequence was loaded into a Bioseq object and the DNase coverage tracks were loaded into Cover objects using the create_from_bam method. Furthermore, we made use of the normalization functionality associated with Cover to perform TPM and z-score of log(count + 1) normalization. Data augmentation for the coverage tracks were achieved randomly flipping the 5′ to 3′ orientation of the tracks using special dataset wrappers that are offered by the Janggu package.

We implemented the architectures given in Supplementary Tables 1, 2 for the individual models using keras and the Janggu model wrapper. The individual submodels were combined by removing the output layer, concatenating the top-most hidden layers and adding a new output layer.

Training was performed using a binary cross-entropy loss with AMSgrad[20] for at most 30 epochs using early stopping monitored on the validation set with a patience of 5 epochs. We trained each model 5 times with random initialization in order to assess reproducibility. Performance were measured on the independent test chromosome using the area under the precision-recall curve (auPRC).

**Dataset and evaluation of published models**. Following the instructions of Zhou et al.[4], we downloaded the human genome hg19 and obtained narrowPeak files for 919 features from ENCODE and ROADMAP from the URLs listed in Supplementary table 1 of Zhou et al.[4] Broken links were adapted where necessary, including for the histone modification features.

Training, validation and test regions were obtained from http://deepsea.princeton.edu/(allTFs.pos.bed.tar.gz). Following Zhou et al.[4], all regions on chromosomes 8 and 9 were assigned to the test set. In contrast to the original training-validation set split of (2,200,000 training, 4000 validation samples), we opted for a more conservative 90%/10% training-validation split to reduce the number of features with no positive examples in the validation set, since we wanted to utilize the benchmark to test different model variants.

We implemented the model architectures described in Zhou et al.[4] and Quang et al.[17] using keras and the Janggu model wrapper. In addition, the models were adapted to scan both DNA strands rather than only the forward strand using the DnaConv2D layer, available in the Janggu library.

For training and evaluation, we served up the model with sequences and output labels that were loaded as Bioseq and Cover objects from Janggu. We compared different context window sizes 500 bp, 1000 bp, and 2000 bp as well as mono-, di- and tri-nucleotide based sequence encoding.

The models were trained using AMSgrad[20] for at most 30 epochs using early stopping with a patience of 5 epochs.

We evaluated the performance using the auPRC on the independent test regions.

**Dataset and evaluation for CAGE-tag prediction**. For the CAGE-tag prediction we focused on human HepG2 cells. We downloaded samples for CAGE (ENCFF177HHM, bam-format), DNase (ENCFF591XCX, bam-format) and H3K4me3 (ENCFF736LHE, bigWig format) from the ENCODE project. Moreover, we used the hg38 reference genome and extracted the set of all protein coding gene promoter regions (200 bp upstream from the TSS) from GENCODE version V29 which constitute the ROI.

We loaded the DNA sequence using a ±350 bp flanking window using the Bioseq object. For CAGE, DNase and H3K4me3, we summed the signal for each promoter using flanking windows of 400 bp, 200 bp, and 200 bp to each dataset, respectively. The promoter signals for each feature were subsequently log-transformed using a pseudo-count of one and then Z score normalized. The coverage data were extracted and transformed using the create_from_bigwig and create_from_bam constructors of the Cover object.

The DNA and chromatin-based models are summarized in Supplementary Tables 3, 4. They were implemented using keras and the Janggu model wrapper. Furthermore, the joint model is built by concatenating the top most hidden layers and adding a new output layer. We pursued a cross-validation strategies where we

trained a model on genes of all chromosomes but one validation autosome, repeating the process for each autosome. Genes on chromosome 1 were left out entirely from the cross-validation runs and were used for the final evaluation. The models were trained using mean absolute error loss with AMSgrad[20] for at most 100 epochs using early stopping with a patience of 5 epochs.

For the evaluation of the model performance, we used the Pearson's correlation between the predicted and observed CAGE signal on the test dataset.

**Reporting summary**. Further information on research design is available in the Nature Research Reporting Summary linked to this article.

## Data availability

All datasets used in this study are publicly available.

For use case 1 we obtained the following ENCODE and ROADMAP datasets https://www.encodeproject.org/files/ENCFF446WOD/@@download/ENCFF446WOD.bed.gz, https://www.encodeproject.org/files/ENCFF546PJU/@@download/ENCFF546PJU.bam, https://www.encodeproject.org/files/ENCFF059BEU/@@download/ENCFF059BEU.bam. Blacklisted regions were obtained from http://mitra.stanford.edu/kundaje/akundaje/release/blacklists/hg38-human/hg38.blacklist.bed.gz. The human genome version hg38 was obtained from http://hgdownload.cse.ucsc.edu/goldenPath/hg38/bigZips/hg38.fa.gz.

For use case 2 we used the set of narrowPeak files summarized in https://github.com/wkopp/janggu_usecases/tree/master/extra/urls.txt (archived version v1.0.1). The human genome version hg19 was obtained from http://hgdownload.cse.ucsc.edu/goldenPath/hg19/bigZips/hg19.fa.gz

For use case 3 we used the ENCODE datasets https://www.encodeproject.org/files/ENCFF591XCX/@@download/ENCFF591XCX.bam, https://www.encodeproject.org/files/ENCFF736LHE/@@download/ENCFF736LHE.bigWig, https://www.encodeproject.org/files/ENCFF177HHM/@@download/ENCFF177HHM.bam as we as the GENCODE annotation v29 from ftp://ftp.ebi.ac.uk/pub/databases/gencode/Gencode_human/release_29/gencode.v29.annotation.gtf.gz.

## Code availability

Janggu is freely available using the pypi echosystem and via github under a GPL-v3 license at https://github.com/BIMSBbioinfo/janggu A comprehensive documentation, including tutorials, can be found at https://janggu.readthedocs.io. Code for reproducing the use cases is provided on github: https://github.com/wkopp/janggu_usecases.

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

## Acknowledgements

The authors wish to thank Jonathan Ronen for valuable comments on the manuscript. W.K. was supported by the German Federal Ministry of Education and Research (de. NBI; FKZ 031L0101B). U.O. acknowledges funding from BMBF project MechML.

## Author contributions

W.K. designed and implemented Janggu with input from A.A. A.T. contributed to library development. W.K. and R.M. performed data analysis for the use cases. U.O. helped with the use-case concept. A.A. supervised the work and organized the resources dedicated to the project. All authors contributed to manuscript writing.

## Competing interests

The authors declare no competing interests.
