## [Peer Review File · Nature Communications]

Reviewers' comments:

Reviewer #1 (Remarks to the Author):

Summary

In this work, the authors present a new software package, Janggu, that supports deep learning in genomics. Janggu allows users to quickly convert a wide variety of genomics data file formats to data objects compatible with keras and scikit-learn. Janggu provides a keras wrapper for users to easily train and evaluate sequence-based deep learning models.

The library is well documented and the authors provide many examples (in the Github repository and website) of how to use Janggu in practice.

Major comments:

1. Currently, the case studies in the paper read like research results; it would be more helpful if each case study explicitly stated the functionality from Janggu that was used. A large part of the paper is focused on how higher order sequence encodings can improve performance in the selected case studies. For instance, the first case study spends multiple paragraphs discussing how dropout improves performance for higher order sequence encodings, even though dropout is an aspect of model architecture design that has nothing to do with the library. In all of the case studies, it is unclear to me which steps directly use Janggu and which do not.

2. I do not find their claim that Janggu is "directly compatible with popular deep learning libraries" outside of Keras to be accurate. The authors mention that Janggu differs from Selene in that Selene is tightly integrated with a specific neural network library (PyTorch)--but Janggu is similarly integrated with Keras. That is, even though dataset objects produced by Janggu can be used with other deep learning libraries as well as machine learning libraries like scikit-learn, I cannot directly use the training and evaluation components of Janggu to train a model specified in PyTorch. Certainly, the ability to use Janggu quickly and easily with Keras is a useful new contribution, but I find the flexibility in data processing to be the primary contribution of this library. What Janggu has that sets it apart from the existing genomics deep learning libraries is the flexibility with which users can generate training data from diverse genomics data types. Figure 1 seems to emphasize this fact as well, as the largest part of the figure is the different kinds of data that the library supports. The authors should adjust the language surrounding the library (e.g. in the abstract, introduction/related work) to reflect this.

3. Case study 2:

* Could the authors explain in the main text why they trained both DeepSEA and DanQ for this case study? For example, the authors might say that the 2 models were trained to show that Janggu can be used to easily compare 2 models. Otherwise, it seems unnecessary to include DeepSEA at all, since Fig. 3 only focuses on results from DanQ.

* The authors should also note in the main text that DanQ outperforms DeepSEA specifically for auPRC.

* Model performance depends strongly on both the training hyperparameters and model architecture. Since they retrained DeepSEA and DanQ with different hyperparameters (i.e. AMSgrad and limiting the number of epochs) from the original models, it is important for the authors to clarify in the text/figures that this is the case. The comparison made in this case study is very different from the comparison done in the DanQ manuscript, so the sentence citing the DanQ manuscript (i.e. "as reported previously") should be revised.

4. Are the case studies in the manuscript provided in the `examples` directory of the GitHub repository? If not, is it possible to provide that code and/or point to what examples are most

similar to those described in the paper? In general, it would be very helpful to link to the examples and provide some description/overview of each example in the main GitHub README.

Minor comments:

5. The authors write that they expect that Janggu will offer “significant reductions in repetitive software engineering aspects” associated with pre-processing steps. Can they illustrate this in their case studies? For example, how easy is it to generate all the different datasets for the JunD binding prediction example? How much work would it have been without Janggu? The legend for Fig. 2 should indicate that the different one-hot encoding orders and normalization methods are all supported by Janggu.

6. Related to the previous point: with the exception of Fig. 1, the figures do not reference the library at all. Without looking at Fig. 1, I would have no idea that this paper was about a software library.

7. Please make Fig. 3A and B easier to read. Also, is it possible to adjust the scales for subfigures C-E so that they are consistent with each other, or note in the text that there is a significant difference in scale between TFs and DNase/Histone?

8. I find that the title/framing of Janggu in this paper somewhat limits the use case of the library. Janggu supports machine learning in genomics, with added support for deep learning through keras.

9. The authors conclude that these use cases “confirmed that higher-order sequence features improve deep learning models.” They go further and say that they expect “improved accuracies for variant effect predictions” with higher-order encodings even though they have not done this evaluation. These statements may be too strong for the scope of this paper. The examples that the authors use for this paper are intended to illustrate Janggu. Even if their claim about performance improvements (by auPRC, etc.) for higher-order sequence encodings is true in these contexts, it should not be the focus of the manuscript.

Reviewer #2 (Remarks to the Author):

Most deep learning tools were designed to address a specific question on a fixed data set or by a fixed model architecture. In this paper, the authors introduced Janggu, a python library that allows easy access to common genomics data formats and provided useful tools for model evaluation. It piggybacks on existing python deep learning packages like Keras to utilize their strength. The authors demonstrate the usefulness of Janggu via three deep learning genomics applications.

Overall, I think Janggu is a mature python implementation and would be helpful to users who wish to apply deep learning to genomics data. Its improvements over existing packages like pysster, kipoi, and selene were also well explained. I believe the computational biology community should welcome this package.

Some comments below:

Major:

- I could not find the supplementary materials (both in the review system and on biorxiv).
- For the second experiment, the authors compared their implementations with the original model of DeepSEQ and DanQ. Yet, I did not find any empirical results supporting their claim, both in the

paper and in the Janggu documentation. I suggest the authors add empirical comparisons to the two models that hopefully show some consistency.

- It would be helpful to discuss the scalability of the implementation.

Minor:

- The author may consider making Figures 3C-E horizontal for better visualization.
- Wherever Pearson's correlation is used, it would be nice to provide a p-value, possibly based on linear regression and F-test.

Reviewer #3 (Remarks to the Author):

Kopp et al. present an end-to-end deep learning framework, Janggu, for genomics problems. Janggu accepts a wide range of genomics data formats and can preprocess and normalize data in multiple ways, train models using keras or sklearn, and evaluate and visualize model predictions. A quick inspection of the code and running some examples indicates that it has been properly developed and tested. While I think the proposed method checks many important boxes, this manuscript falls a bit short on demonstrating its capabilities. Please find my comments and rationale below.

Major Comments

- The most important claim that authors make is Janggu makes replicating and retraining genomics models very easy. For example, they retrain DeepSEA and DanQ models. The retrained models must be compared with the original models and the concordance between predictions needs to be tested and verified. This seems more important than DeepSEA vs. DanQ models.

- There is no discussion about hyperparameter tuning in the manuscript or documentations. Performance of deep learning model or even simple elastic nets could significantly vary based on the hyperparameter setting. Since they are using keras, the authors might want to use the `kerastuner` package (<https://github.com/keras-team/keras-tuner>). For sklearn models, they can use `*CV` classes (e.g. ElasticNetCV).

- A major criticism of deep models used to be their black box nature; however, in the past few years several ideas have been proposed such as saliency maps (<https://arxiv.org/abs/1312.6034>) and mutation maps (<https://www.nature.com/articles/nbt.3300>) or DeepLIFT (<https://arxiv.org/abs/1704.02685>) for genomics data. It appears that there is some importance attribution implemented, but this needs to be discussed in the manuscript.

- In the JunD example, the authors claim that using di- and tri-nucleotide encoding improves performance. This is interesting and might indeed be real, but training and testing models on the same data acquisition technology can be problematic. What if the model is just learning ChIP-seq biases and artifacts? This could be a consequence of the way negative examples are defined. There are two things that needs to be shown here:

1. Model's performance when applied to a different technology, e.g., SELEX [PMID: 28473536].
2. Performance of a simple di- or tri-nucleotide composition model should also be reported.

- Data augmentation by randomly flipping strands of inputs seems incorrect. When dealing with double-stranded DNA sequences, forward and reverse strands must be treated exactly the same. That is, model's output for ACGTTA should be exactly the same as TAACGT.

- The authors could elaborate on generating negative examples a bit. There are multiple ways (e.g. dinucleotide composition-preserving shuffling) and they might have drastic effect on the model.

Minor Comments

- Adding ensembles (e.g. bagging) would be very helpful for reducing model variance.

Response to reviewer comments

We are grateful to all three reviewers for their time and effort and wish to thank them for their constructive and helpful comments on the manuscript. We have revised the manuscript accordingly and hope that the changes and our point-by-point response below is adequate.

Response to reviewer comments

Reviewer #1

1. Currently, the case studies in the paper read like research results; it would be more helpful if each case study explicitly stated the functionality from Janggu that was used. A large part of the paper is focused on how higher order sequence encodings can improve performance in the selected case studies. For instance, the first case study spends multiple paragraphs discussing how dropout improves performance for higher order sequence encodings, even though dropout is an aspect of model architecture design that has nothing to do with the library. In all of the case studies, it is unclear to me which steps directly use Janggu and which do not.

Thank you for your suggestion. We have purposely tried to minimize adding too many technical details about which Janggu functions were used in the use case sections to preserve readability of the experiments also for readers with non-software development background.

In all use cases, we made use of Janggu's dataset objects Bioseq and Cover as well as the Janggu model wrapper which are summarized in the hallmark section of the manuscript.

We have deferred the majority of the technical details to the methods section and attempted to further improve the methods section in the revision. We also added a link to the source code for reproducing the use cases (in the online documentation and the manuscript). We hope that this clarifies how Janggu was used for the demonstrations.

2. I do not find their claim that Janggu is “directly compatible with popular deep learning libraries” outside of Keras to be accurate. The authors mention that Janggu differs from Selene in that Selene is tightly integrated with a specific neural network library (PyTorch)--but Janggu is similarly integrated with Keras. That is, even though dataset objects produced by Janggu can be used with other deep learning libraries as well as machine learning libraries like scikit-learn, I cannot directly use the training and evaluation components of Janggu to train a model specified in PyTorch. Certainly, the

ability to use Janggu quickly and easily with Keras is a useful new contribution, but I find the flexibility in data processing to be the primary contribution of this library. What Janggu has that sets it apart from the existing genomics deep learning libraries is the flexibility with which users can generate training data from diverse genomics data types. Figure 1 seems to emphasize this fact as well, as the largest part of the figure is the different kinds of data that the library supports. The authors should adjust the language surrounding the library (e.g. in the abstract, introduction/related work) to reflect this.

While it is true that some parts of Janggu are based on an integration with keras, we would like to stress that the statements referred to by reviewer #1 specifically refer to the dataset objects. For example, in the abstract we specifically note: "Through a numpy-like interface, **the dataset objects** are directly compatible with popular deep learning libraries, including keras." Afterwards, in the introduction we state "**These objects** [the dataset objects] provide easy access ..., and they are directly compatible with commonly used deep learning libraries, such as keras or scikit-learn."

However, in order to avoid any confusion, we adjusted the hallmarks sections to explicitly mention that some features of Janggu are based on a keras integration.

3. Case study 2:

*** Could the authors explain in the main text why they trained both DeepSEA and DanQ for this case study? For example, the authors might say that the 2 models were trained to show that Janggu can be used to easily compare 2 models. Otherwise, it seems unnecessary to include DeepSEA at all, since Fig. 3 only focuses on results from DanQ.**

We included both models to be able to compare them. In this case, we can see for example that DanQ outperforms DeepSEA on our benchmark analysis (see Figure S.1).

*** The authors should also note in the main text that DanQ outperforms DeepSEA specifically for auPRC.**

We believe that we have clearly described that we use the auPRC as a performance metric at various occasions for this and other use cases, including in the methods section and the figure and we have also noted that DanQ outperforms DeepSEA.

*** Model performance depends strongly on both the training hyperparameters and model architecture. Since they retrained DeepSEA and DanQ with different hyperparameters (i.e. AMSgrad and limiting the number of epochs) from the original models, it is important for the authors to clarify in the text/figures that this is the case. The comparison made in this case study is very different from the comparison done in the DanQ manuscript, so the sentence citing the DanQ manuscript (i.e. "as reported previously") should be revised.**

It is true that we have not performed the comparison between DanQ and DeepSEQ identically to the original publication, but we took the liberty of selecting slightly different hyperparameters as well as used an adapted benchmark dataset.

While it is possible to choose different hyperparameters and even to optimize the hyperparameter choices, this was not our primary goal.

Our primary goal is to demonstrate the flexibility and range of applications Janggu can be applied to, including utilizing a published network architecture and experimenting with new aspects (e.g. higher-order sequence encoding or different flanking windows).

We documented our training and evaluation setup in the methods section of the manuscript and attempted to improve the methods section for the revision. Furthermore, we included a link to the repository containing the use cases which enables interested users to experiment with other aspects, including using different optimizers.

We used AMSgrad (a derivative of ADAM) in all use cases because it is widely considered as a robust optimizer choice and we have used early stopping with each use case.

Finally, we would like to point out that our comparison is not very different from the comparison done in the DanQ paper. Even though we have used a slightly different training setup, in essence we have compared the DeepSEA and the DanQ model in our benchmark analysis and find that DanQ consistently outperforms DeepSEA. This general tendency of DanQ outperforming DeepSEA was also reported in the DanQ paper. We edited the text to show more clearly how benchmarks are done as stated above.

Minor comments:

5. The authors write that they expect that Janggu will offer “significant reductions in repetitive software engineering aspects” associated with pre-processing steps. Can they illustrate this in their case studies? For example, how easy is it to generate all the different datasets for the JunD binding prediction example? How much work would it have been without Janggu? The legend for Fig. 2 should indicate that the different one-hot encoding orders and normalization methods are all supported by Janggu.

As we pointed out in the manuscript, among the main benefits of Janggu are the dataset objects, which help to reduce redundant source code. Data acquisition in genomics for deep learning applications is a major bottleneck, which consumes time and effort. However, using Janggu, data can be easily loaded and parametrize the dataset objects. Thus, the developers can focus on biological hypothesis testing rather spending too much time on the technical overhead associated with the application. To demonstrate this, in this and the other use cases, we use Janggu with different file formats, model architecture as well as classification and regression tasks. A range of additional examples are available in the online documentation.

We added notes in the caption of Figure 2 to indicate that different one-hot encoding orders and normalization options are available with the Bioseq and Cover objects. These aspects were also

highlighted in the hallmarks section of the manuscript. We also attempted to improve the methods sections to clarify the use of these features.

6. Related to the previous point: with the exception of Fig. 1, the figures do not reference the library at all. Without looking at Fig. 1, I would have no idea that this paper was about a software library.

Fig 1 and the hallmark section describe the scope of the library. The remaining figures illustrate example applications where we made use of Janggu's functionality. However, to maintain readability for non-programmers, we deferred the technical details to the methods section.

7. Please make Fig. 3A and B easier to read. Also, is it possible to adjust the scales for subfigures C-E so that they are consistent with each other, or note in the text that there is a significant difference in scale between TFs and DNase/Histone?

We removed the text labels from 3A in line with the reviewer's suggestion for better readability, however, for 3B we left the labels, because they are informative about which TFs benefit most from the higher order sequence encoding. Finally, we replaced Figure 3C-E with a boxplot figure to improve readability.

8. I find that the title/framing of Janggu in this paper somewhat limits the use case of the library. Janggu supports machine learning in genomics, with added support for deep learning through keras.

Thank you for your suggestion. While we agree that some aspects can be used for machine learning models other than neural networks, parts of the library including the importance attribution or some model evaluation aspects require the use of keras and are therefore specifically targeted for deep learning applications. Therefore, our main focus rest on deep learning applications. Depending on the users' needs and future developments, we might reconsider some of the Janggu's functionality for other machine learning applications. However, this is currently beyond the scope of the package.

9. The authors conclude that these use cases “confirmed that higher-order sequence features improve deep learning models.” They go further and say that they expect “improved accuracies for variant effect predictions” with higher-order encodings even though they have not done this evaluation. These statements may be too strong for the scope of this paper. The examples that the authors use for this paper are intended to illustrate Janggu. Even if their claim about performance improvements (by auPRC, etc.) for higher-order sequence encodings is true in these contexts, it should not be the focus of the manuscript.

In line with the reviewer's suggestion, we removed the statement about the improved accuracies for variant effect predictions in the conclusion part, but we added this point to the discussion

section explaining that the improved model accuracies (based on the higher-order sequence features) will benefit the variant effect predictions, because they depend directly on the model predictions.

Reviewer #2

Major:

- I could not find the supplementary materials (both in the review system and on biorxiv).

It is correct that on the review system, the supplementary notes were not attached. We apologize for that. The supplementary notes are however included on biorxiv (see the last three pages of the pdf).

The revision contains the supplementary notes which summarize the neural network architectures used for the case studies as well as a performance comparison for the DeepSEA and DanQ use case (Fig S.1).

- For the second experiment, the authors compared their implementations with the original model of DeepSEQ and DanQ. Yet, I did not find any empirical results supporting their claim, both in the paper and in the Janggu documentation. I suggest the authors add empirical comparisons to the two models that hopefully show some consistency.

Assuming that by empirical comparison the reviewer is asking to compare our AUC score with the published AUC score from Zhou et. al, we have not included these for several reasons: First and foremost, the benchmark datasets are not identical in our case compared to the original datasets. Consequently, a direct comparison of the AUC's is not justified.

There are two reasons for the differences between the datasets:

- 1) We have selected a larger validation set compared than was used Zhou et. al. While Zhou et. al have allocated 2200000 sequences for training and 4000 for validation, we opted for a more conservative 90%/10% split between training and validations set as we wanted to proceed with new experiments (different flanking regions and orders). We made this choice, because we noticed that for some TFs no binding events occur in the original validation set of 4000 data points and the use of a larger validation set alleviates this issue. As a consequence, this leaves fewer data points for model fitting, which might slightly decrease the performances. Such performance losses are of no concern for us, since our analysis compares the models on our own fixed benchmark analysis. However, this would create an unfair situation when comparing the AUC's across the different benchmark datasets.
- 2) Another reason for the differences between the benchmark datasets is that some of the original urls in supplementary table 1. of Zhou et. al. do not exist anymore, including the histone modifications. As we pointed out in the methods section, we substitute these files with files that were downloaded from other sources, which are not necessarily identical to the original source. In fact, we found evidence that the peaks for the histone modifications are not identical with the histone modification peaks that were originally

used. Consequently, this adds to the differences between the benchmark datasets and would render a direct comparison between AUC's inappropriate.

In the revised manuscript, we attempted to further improve the methods section in order to highlight the differences between the benchmark datasets.

In addition, we added a link to the repository containing the use cases to the manuscript to facilitate reproducibility.

- It would be helpful to discuss the scalability of the implementation.

We have attempted to address the aspect of scalability in the dataset objects by offering different options of how the data is stored internally. These can be as 1) numpy array, 2) sparse array or 3) as hdf5 dataset. These options allow to balance between memory requirements and speed depending on the available computing infrastructure. For example, numpy arrays can be consumed very fast, but they might be too large to be stored in the process's memory. Large datasets may be consumed from disk with the hdf5 format option. The sparse array is convenient for sparse datasets, including peak calls.

This aspect was briefly outlined in the hallmarks section of the manuscript and it is also described in the online documentation.

Minor:

- The author may consider making Figures 3C-E horizontal for better visualization.

In line with the reviewer's request, we replaced Fig 3C-E with boxplots for better visibility.

- Wherever Pearson's correlation is used, it would be nice to provide a p-value, possibly based on linear regression and F-test.

We have added a regression line based on linear regression to Fig 4 along with the F-test statistics and a P-value. We would also like to point out that Table 1 already summarizes the Pearson's correlation coefficients with estimates of the standard error which indicates that the confidence intervals are very narrow and as a consequence results in extremely small P-values ($< 2.2e-16$). This is in part due to the fact that the test set contains >3000 data points.

Reviewer #3

Major Comments

- The most important claim that authors make is Janggu makes replicating and retraining genomics models very easy. For example, they retrain DeepSEA and DanQ models. The retrained models must be compared with the original models and the concordance

between predictions needs to be tested and verified. This seems more important than DeepSEA vs. DanQ models.

We have not included a comparison with the AUC score from Zhou et. al for several reasons: First and foremost, the benchmark datasets are not identical in our case compared to the original datasets. Consequently, a direct comparison of the AUCs is not justified.

There are two reasons for the differences between the datasets:

- 1) We have selected a larger validation set compared than was used Zhou et. al. While Zhou et. al have allocated 2200000 sequences for training and 4000 for validation, we opted for a more conservative 90%/10% split between training and validations set as we wanted to proceed with new experiments (different flanking regions and orders). We made this choice, because we noticed that for some TFs no binding events occur in the original validation set of 4000 data points and the use of a larger validation set alleviates this issue. As a consequence, this leaves fewer data points for model fitting, which might slightly decrease the performances. Such performance losses are of no concern for us, since our analysis compares the models on our own fixed benchmark analysis. However, this would create an unfair situation when comparing the AUC's across the different benchmark datasets.
- 2) Another reason for the differences between the benchmark datasets is that some of the original urls in supplementary table 1. of Zhou et. al. do not exist anymore, including the histone modifications. As we pointed out in the methods section, we substitute these files with files that were downloaded from other sources, which are not necessarily identical to the original source. In fact, we found evidence that the peaks for the histone modifications are not identical with the histone modification peaks that were originally used. Consequently, this adds to the differences between the benchmark datasets and would render a direct comparison between AUC's inappropriate.

We would like to emphasize that our goal for this use case is not to demonstrate that we can precisely reproduce the original performances, but rather do we want to illustrate that Janggu allows to effectively reimplement published model architectures from scratch using raw genomic files in order to reuse, repurpose them for asking different questions, or simply experimenting with different variants of the network.

In the revised manuscript, we attempted to further improve the methods section in order to highlight the differences between the benchmark datasets. We would like to point out that we made the effort to compare different methods for our benchmark dataset even though AUC comparison with the numbers in the original Zhou et. al. paper is impossible. This is an acceptable approach given the limitations itemized above.

In addition, we added a link to the repository containing the use cases to the manuscript to facilitate reproducibility.

- There is no discussion about hyperparameter tuning in the manuscript or documentations. Performance of deep learning model or even simple elastic nets could significantly vary based on the hyperparameter setting. Since they are using keras, the authors might want to use the `kerastuner` package (<https://github.com/keras-team/keras-tuner>). For sklearn models, they can use `*CV` classes (e.g. ElasticNetCV).

We agree that hyperparameter optimization is an important component in deep learning modelling, however, it depends on the context, the particular question and the user's interest which parameters ought to be subjected to hyperparameter optimization. A general discussion on hyperparameter optimization is beyond the scope of our manuscript, as our primary goal is to demonstrate that Janggu facilitates easy access to genomics data for deep learning and helps with model evaluation aspects.

However, we have added an example notebook to the online tutorial section on how to use Janggu in combination with hyperparameter optimization employing the hyperopt package for the interested user.

- A major criticism of deep models used to be their black box nature; however, in the past few years several ideas have been proposed such as saliency maps (<https://arxiv.org/abs/1312.6034>) and mutation maps (<https://www.nature.com/articles/nbt.3300>) or DeepLIFT (<https://arxiv.org/abs/1704.02685>) for genomics data. It appears that there is some importance attribution implemented, but this needs to be discussed in the manuscript.

We have mentioned the importance attribution in the hallmarks section of the manuscript, which summarizes Janggu's functionality. Furthermore, we have used this feature for the JunD use case (see Fig. 2D) to highlight that that one of the most important sequence features that is revealed resembles the known JunD motif. The importance attribution is also mentioned in the discussion of the manuscript.

- In the JunD example, the authors claim that using di- and tri-nucleotide encoding improves performance. This is interesting and might indeed be real, but training and testing models on the same data acquisition technology can be problematic. What if the model is just learning ChIP-seq biases and artifacts? This could be a consequence of the way negative examples are defined. There are two things that needs to be shown here:

- 1. Model's performance when applied to a different technology, e.g., SELEX [PMID: 28473536].**
- 2. Performance of a simple di- or tri-nucleotide composition model should also be reported.**

We acknowledge the fact that ChIP-seq data might be biased. However, a follow up investigation of the JunD use case is beyond the scope of this paper, because we are primarily concerned with showing that Janggu can be used to address a wide range of questions using genomics data and deep learning models.

Regarding the JunD use case, while the models may capture ChIP-seq specific biases to some extent, inspection of the results, suggests that the models also capture useful biological signal. For example, the importance attribution feature highlights sequences stretches which highly agree with the known JunD motif (see Fig 2D). Therefore, it is unlikely that exclusively artifacts were picked up.

Furthermore, we would like to stress that we have applied the higher order sequence encoding feature to all use cases and we observe moderate and sometimes substantial performance improvements not only when predicting JunD binding, but also with other benchmark dataset and model types (see use case 2 and to a lesser extend use case 3).

Finally, it is widely accepted that higher order nucleotide composition plays an important role for describing transcription factor binding. In the discussion section we reference a number of publications along that line, including Keilwagen et al 2015.

- Data augmentation by randomly flipping strands of inputs seems incorrect. When dealing with double-stranded DNA sequences, forward and reverse strands must be treated exactly the same. That is, model's output for ACGTTA should be exactly the same as TAACGT.

We believe this is a misunderstanding. The manuscript states that the orientation of coverage tracks are flipped not the DNA sequence. More clearly, We did not flip the strands for the DNA sequence. We flipped the orientation of the coverage tracks in this example.

- The authors could elaborate on generating negative examples a bit. There are multiple ways (e.g. dinucleotide composition-preserving shuffling) and they might have drastic effect on the model.

For simplicity, we chose the flanking 10kb region around peaks as negative set (as long as they didn't overlap any other peak). This was described in the methods part.

While other choices for generating the negative samples are possible, our main goal is to demonstrate the flexibility of the Janggu package by employing it on a broad range of applications. Therefore, optimizing the negative set for use case 1 is beyond the scope of this work.

Minor Comments

- Adding ensembles (e.g. bagging) would be very helpful for reducing model variance.

It may be the case that ensembles help to reduce the model variance, but exploring this option is beyond the scope of this manuscript. Our main purpose is to demonstrate the flexibility of Janggu for addressing a range of questions in genomics using deep learning, since the library is our main contribution in this work.

Reviewers' comments:

Reviewer #1 (Remarks to the Author):

Summary

The authors do address many of the comments made in the previous review: the adjustments they have made to the language in the text and improvements in the figures have helped to better demonstrate the functionality provided in Janggu. That said, there are two main issues that remain inadequately addressed: (1) adjusting the way that Janggu is framed and how Janggu is described against related work such as Selene, and (2) the way that the 2nd case study, about DanQ and DeepSEA, is presented.

Comments

1. The authors continually emphasize in their response to reviewers that the "main benefits of Janggu are the dataset objects, which help to reduce redundant source code." I fully agree with this statement and find Janggu to be an important contribution because of this and the high quality of the authors' code, documentation, and tutorials. However, I do not find the language they currently use in their paper to reflect this sentiment; rather, the authors seem to focus too much on trying to position themselves as a novel deep learning library for genomics. In their response to my first review, the authors mention that they are careful to tie the part about being "directly compatible with popular deep learning libraries" with only the dataset objects portion of Janggu. This is true, and I appreciate their adjustment to Hallmarks to explicitly describe the functionality in Janggu that is integrated with keras. Even so, there are many instances throughout the paper where the deep learning aspect of Janggu and its dependence on keras is understated. My comment #2 is one of the larger examples I take issue with, but here I will list a few other instances:

* In the abstract: "Through a numpy-like interface, the dataset objects are directly compatible with popular deep learning libraries, including keras." This can be adjusted to more explicitly state that "the dataset objects are directly compatible with popular deep learning libraries, and Janggu has added support for model development in keras."

* "Janggu additionally facilitates utilities to monitor the training, performance evaluation and interpretation." This can be adjusted to "... monitor the training ... and interpretation in keras."

* "[the dataset objects] are directly compatible with commonly used deep learning libraries, such as keras or scikit-learn." This can be adjusted by replacing "deep learning" with "machine learning." scikit-learn is not a commonly used deep learning library in genomics.

2. In the related work section of Background, the authors imply that one major way that Janggu differs from Selene is in that Selene has a "tight integration" with "a specific neural network library." The authors responded to my initial comment by saying that they emphasize clearly throughout the text that it is only the dataset objects that are directly compatible with multiple deep learning libraries--I agree with this and appreciate the additions they made to the Hallmarks section. Still, I must maintain that this phrase comparing Selene and Janggu in the related work section misrepresents Janggu, for these reasons:

* It is still unclear to me how the authors make the distinction between Selene having a "tight integration" with PyTorch and Janggu having a looser integration with keras. Both Selene and Janggu list deep learning libraries as installation dependencies, and both include code that specifically facilitates training of models with a single deep learning library. It seems to me the distinction is only made in terms of what "selling points" the two different libraries want to emphasize.

* In both libraries, the functionality for generating datasets and for training models is quite separate (i.e. different modules, different classes/objects initialized). The APIs for both libraries (i.e. Selene's sampler module) can be used to generate datasets that can be saved/used with other deep learning libraries. This means that the out-of-the-box deep learning functionality provided by Janggu is, like Selene, written for a specific library.

All of this is to say: such phrasing reinforces the idea that Janggu is a library that can be easily used with many popular deep learning libraries. Could the authors either provide code examples in their GitHub repository to demonstrate how Janggu can be used with libraries other than keras, or remove/adjust the phrase in the related work section?

3. DeepSEA vs. DanQ case study:

* Could the authors adjust the text here to mention auPRC? For example: "we find that the DanQ model consistently outperforms the DeepSEA model (by auPRC) in our benchmark analysis..." Even if auPRC is mentioned in the associated figure and in other case studies, it is important to be clear about it in this section as well.

* The authors emphasize in the response to reviewers that their "goal for this use case" is to "illustrate that Janggu allows [users] to effectively reimplement published model architectures from scratch using raw genomic files in order to reuse, repurpose ... or simply [experiment]." The problem here is that what the author is trying to convey to a reviewer does not match with what the authors choose to highlight in their main text (the whole first half of the text is about DeepSEA and DanQ) and the corresponding figure (Fig. 3), which focuses on DanQ model performance. Because Fig. 3 does not show any comparison with DeepSEA and DanQ anyway, I recommend the authors remove the model comparison from the text and just highlight the successful training and evaluation of many different variations of the DanQ model. The claim about model comparison distracts from the main point of the text. If the authors want to keep the DeepSEA data processing and training in their Github repository, they might just make a note about this in the case study or methods section (i.e. "To demonstrate generality we have also trained DeepSEA, see <link>").

Reviewer #2 (Remarks to the Author):

I thank the authors for their time and effort to address my comments. I do not have further concerns.

Martin Jinye Zhang

Reviewer #3 (Remarks to the Author):

I thank the authors for taking the time to answer my and other reviewers' questions. After carefully reading the rebuttal, I do not think I can make the decision and defer to the Editor. On one hand, Janggu is implemented solidly and appears to have good performance. On the other hand, the manuscript reads like a (well-written) technical report. It is not as technically detailed and thorough as a protocols article, nor does it provide insight into the learned models and delve into aspects of model training. For example, we see that using higher-order sequence features – arguably the most important contribution of Janggu – Improves performance. Is this improvement meaningful? Can it be replicated in different technologies? There are several other points raised by the reviewers (e.g. hyperparameter tuning) that I believe are crucial for a "deep learning for genomics" library.

Response to reviewer comments

We are grateful to all reviewers for their time and effort and wish to thank them for their constructive and helpful comments on the manuscript. We have revised the manuscript accordingly and hope that the changes and our point-by-point response below are adequate.

Response to reviewer comments

Reviewer #1

Comments

1. The authors continually emphasize in their response to reviewers that the “main benefits of Janggu are the dataset objects, which help to reduce redundant source code.” I fully agree with this statement and find Janggu to be an important contribution because of this and the high quality of the authors' code, documentation, and tutorials. However, I do not find the language they currently use in their paper to reflect this sentiment; rather, the authors seem to focus too much on trying to position themselves as a novel deep learning library for genomics. In their response to my first review, the authors mention that they are careful to tie the part about being “directly compatible with popular deep learning libraries” with only the dataset objects portion of Janggu. This is true, and I appreciate their adjustment to Hallmarks to explicitly describe the functionality in Janggu that is integrated with keras. Even so, there are many instances throughout the paper where the deep learning aspect of Janggu and its dependence on keras is understated. My comment #2 is one of the larger examples I take issue with, but here I will list a few other instances:

Thank you for the suggestion. We adapted the text according to the reviewer's suggestion so as to avoid confusion about which elements of Janggu are independent of the machine learning libraries and which parts depend on keras-based models. Also in the beginning of the Janggu online documentation below the graphical abstract, we emphasize the “out-of-the-box evaluation (for keras models specifically)”.

*** In the abstract: “Through a numpy-like interface, the dataset objects are directly compatible with popular deep learning libraries, including keras.” This can be adjusted to more explicitly state that “the dataset objects are directly compatible with popular deep learning libraries, and Janggu has added support for model development in keras.”**

We modified the abstract accordingly.

*** “Janggu additionally facilitates utilities to monitor the training, performance evaluation and interpretation.” This can be adjusted to “... monitor the training ... and interpretation in keras.”**

We modified this in line with the reviewer’s suggestion.

*** “[the dataset objects] are directly compatible with commonly used deep learning libraries, such as keras or scikit-learn.” This can be adjusted by replacing “deep learning” with “machine learning.” scikit-learn is not a commonly used deep learning library in genomics.**

We modified this according to the reviewer’s suggestion.

2. In the related work section of Background, the authors imply that one major way that Janggu differs from Selene is in that Selene has a “tight integration” with “a specific neural network library.” The authors responded to my initial comment by saying that they emphasize clearly throughout the text that it is only the dataset objects that are directly compatible with multiple deep learning libraries--I agree with this and appreciate the additions they made to the Hallmarks section. Still, I must maintain that this phrase comparing Selene and Janggu in the related work section misrepresents Janggu, for these reasons:

*** It is still unclear to me how the authors make the distinction between Selene having a “tight integration” with PyTorch and Janggu having a looser integration with keras. Both Selene and Janggu list deep learning libraries as installation dependencies, and both include code that specifically facilitates training of models with a single deep learning library. It seems to me the distinction is only made in terms of what “selling points” the two different libraries want to emphasize.**

The “tight integration” is only one aspect distinguishing Selene and Janggu. Another important aspect relates to the types of models that can easily be addressed. Both of them were referenced in the main text.

While there are arguably similarities between Selene and Janggu, there are a number of differences between Janggu and Selene making either library more easily applicable in certain situations than its counterpart. Selene’s most important asset is the command-line interface which allows to train, evaluate and analyse models based on a pre-specified configuration with limited programming effort. However, the easy handling through a configuration places restrictions on the modeling flexibility, not only by requiring modeling in pytorch but also regarding the types of models that can be addressed easily. On the other hand, the most

important asset of Janggu, as the reviewer has appreciated, is the dataset objects which enable experimenting with a broad range of models.

We attempted to demonstrate this in our use cases and online tutorial, some of which cannot easily be implemented with Selene without substantial effort:

1. Selene offers limited support for genomics file formats, apart from fasta files and bed files when used for classification-tasks. This means in many cases where the modeling deviates from the DeepSEA-type of models (DNA/Protein sequence as input -> classification labels as output), the users need to perform preprocessing themselves. This is exemplified in the Selene tutorial for a regression task (see regression_mpra_example tutorial in the selene github repository) where the user has to extract and perform preprocessing of the relevant features beforehand. Likewise, since there is no support for coverage data, e.g. from BAM or BIGWIG files, the burden of preprocessing and feature extraction lies with the user. By contrast, the Janggu datasets can readily be used for such tasks. For example, in the use cases of our manuscript, we utilize data sets from different file formats, including FASTA, BAM, BIGWIG and BED format. We also illustrate classification and regression-task scenarios.
2. Selene's support focusses on single-modal models for classification tasks. In particular, models that take the DNA/Protein sequence as input. On the other hand, multi-modal models are currently not supported out-of-the-box. That is models that take multiple input features e.g. DNA and coverage. Likewise, architectures using auxiliary outputs are not supported out-of-the-box. Custom solutions would need to be implemented by the user, requiring programming proficiency and a deep understanding of the selene-sdk library, since the use cases primarily focus on how to set up the configuration file for running Selene from the command line interface, rather than demonstrating how to utilize the selene-sdk directly. By contrast, in our use cases we illustrate single- as well as multi-modal models, e.g. where DNA and DNase coverage is used as input for the model simultaneously. Thereby we demonstrate that Janggu can easily be used to implement a range of model types.
3. While Selene's command-line interface only works with pytorch, some components of the library may well be used for custom models that e.g. use keras models. However, to our knowledge, this has not been demonstrated before (e.g. on the online tutorial or in the paper) and therefore would likely require a fair amount of programming proficiency. On the other hand, our tutorial provides explicit examples on how to use Janggu datasets with various machine learning libraries, including keras, pytorch and scikit-learn. **(The pytorch example has been added since the last revision).**

In summary, Selene and Janggu have their individual strengths and weaknesses. Selene's configurable command-line interface allows setting up deep learning applications for certain

models with little programming effort, whereas Janggu offers greater modeling flexibility through the dataset objects.

We agree that there is additional support in Janggu for keras based models, including through the reverse complement layer. But we hope to have clarified the dependence on keras for certain features (see our answer above point). We also hope to have clarified the difference between Selene and Janggu and that this is not just based on integrating with a different deep learning library.

*** In both libraries, the functionality for generating datasets and for training models is quite separate (i.e. different modules, different classes/objects initialized). The APIs for both libraries (i.e. Selene's sampler module) can be used to generate datasets that can be saved/used with other deep learning libraries. This means that the out-of-the-box deep learning functionality provided by Janggu is, like Selene, written for a specific library.**

Please see our answer above, regarding the differences between the libraries.

Accordingly, the question about what entails "out-of-the-box" support does not only rely on the deep learning library that is used, but also which types of models are best supported.

As mentioned above, both libraries have their strengths and weaknesses in certain aspects.

All of this is to say: such phrasing reinforces the idea that Janggu is a library that can be easily used with many popular deep learning libraries. Could the authors either provide code examples in their GitHub repository to demonstrate how Janggu can be used with libraries other than keras, or remove/adjust the phrase in the related work section?

We have updated the online tutorial to feature examples on how to use Janggu datasets with different libraries, including keras, pytorch and scikit-learn. The pytorch example was added since the last revision.

3. DeepSEA vs. DanQ case study:

*** Could the authors adjust the text here to mention auPRC? For example: "we find that the DanQ model consistently outperforms the DeepSEA model (by auPRC) in our benchmark analysis..." Even if auPRC is mentioned in the associated figure and in other case studies, it is important to be clear about it in this section as well.**

We added the remark to the auPRC in the text according to the the reviewer's suggestion.

*** The authors emphasize in the response to reviewers that their "goal for this use case" is to "illustrate that Janggu allows [users] to effectively reimplement published model architectures from scratch using raw genomic files in order to reuse, repurpose ... or simply [experiment]." The problem here is that what the author is trying to convey to a reviewer does not match with what the authors choose to highlight in their main text (the whole first half of the text is about DeepSEA and DanQ) and the corresponding figure (Fig. 3), which focuses on DanQ model performance. Because Fig. 3 does not show any**

comparison with DeepSEA and DanQ anyway, I recommend the authors remove the model comparison from the text and just highlight the successful training and evaluation of many different variations of the DanQ model. The claim about model comparison distracts from the main point of the text. If the authors want to keep the DeepSEA data processing and training in their Github repository, they might just make a note about this in the case study or methods section (i.e. "To demonstrate generality we have also trained DeepSEA, see).

We haven't added the comparison between DeepSEA and DanQ in the Fig.3 due to the limited space. However, we have included it as Supplementary Fig. S.1. The results are referenced in the main text which explains that DanQ output performs DeepSEA.

Reviewer #3 (Remarks to the Author):

I thank the authors for taking the time to answer my and other reviewers' questions. After carefully reading the rebuttal, I do not think I can make the decision and defer to the Editor. On one hand, Janggu is implemented solidly and appears to have good performance. On the other hand, the manuscript reads like a (well-written) technical report. It is not as technically detailed and thorough as a protocols article, nor does it provide insight into the learned models and delve into aspects of model training. For example, we see that using higher-order sequence features – arguably the most important contribution of Janggu – Improves performance. Is this improvement meaningful? Can it be replicated in different technologies? There are several other points raised by the reviewers (e.g. hyperparameter tuning) that I believe are crucial for a "deep learning for genomics" library.

We thank the reviewer for the comments and appreciate the positive feedback on the implementation. Many technical details regarding how to use Janggu are in the tutorials of the package documentation website (<https://janggu.readthedocs.io/>). This includes a tutorial on how to use hyperparameter optimization together with Janggu as well as a link to the source code for reproducing the use cases. We made sure that users will have no problem getting started by providing extensive technical documentation. Some of the points raised have been addressed in the earlier round where we mentioned previous studies showing higher-order sequence features are important for transcription factor binding and discussed in the manuscript. Therefore, Janggu provides the capability to include this known feature for TFBS detection in deep learning models. We are not claiming that the higher-order features are a new discovery in TFBS related research.

REVIEWERS' COMMENTS:

Reviewer #1 (Remarks to the Author):

The authors have addressed all my points successfully.